# Mastitis in Dairy Cattle: On-Farm Diagnostics and Future Perspectives

**DOI:** 10.3390/ani13152538

**Published:** 2023-08-06

**Authors:** Chiara Tommasoni, Enrico Fiore, Anastasia Lisuzzo, Matteo Gianesella

**Affiliations:** Department of Animal Medicine, Production and Health, University of Padua, Viale dell’Università 16, 35020 Legnaro, Italy; enrico.fiore@unipd.it (E.F.); anastasia.lisuzzo@phd.unipd.it (A.L.); matteo.gianesella@unipd.it (M.G.)

**Keywords:** mastitis, dairy cattle, diagnosis, control, on-farm culture, mammary ultrasound, blood gas analysis, Infra-Red Thermography, California Mastitis Test, electrical conductivity

## Abstract

**Simple Summary:**

Mastitis is one of the most common diseases of the dairy industry and with it brings important economic losses. The most prevalent form of the disease is subclinical mastitis, which leads, in the absence of clinical signs, to decreased milk production, increased somatic cell count, and an increased risk of clinical mastitis during lactation. With the increasing public health concerns for antimicrobial use and its relationship with the development of antimicrobial resistance, nation-specific regulations and general pressure to reduce group-level prophylactic use of antimicrobials have been established. Selective dry therapy reserves antimicrobial treatment for cows or quarters suspected of having an intramammary infection. The treatment is administered after the last milking, while uninfected cows or quarters do not receive antibiotherapy. Since selective dry cow therapy was introduced, different methods of selecting infected cows or quarters have been reported. The aim of this article is to describe the management of mastitis in dairy cows and the main tools for its diagnosis, with a specific focus on on-farm instruments.

**Abstract:**

Mastitis is one of the most important diseases in dairy cattle farms, and it can affect the health status of the udder and the quantity and quality of milk yielded. The correct management of mastitis is based both on preventive and treatment action. With the increasing concern for antimicrobial resistance, it is strongly recommended to treat only the mammary quarters presenting intramammary infection. For this reason, a timely and accurate diagnosis is fundamental. The possibility to detect and characterize mastitis directly on farm would be very useful to choose the correct management protocol. Some on-field diagnostic tools are already routinely applied to detect mastitis, such as the California Mastitis Test and on-farm culture. Other instruments are emerging to perform a timely diagnosis and to characterize mastitis, such as Infra-Red Thermography, mammary ultrasound evaluation and blood gas analysis, even if their application still needs to be improved. The main purpose of this article is to present an overview of the methods currently used to control, detect, and characterize mastitis in dairy cows, in order to perform a timely diagnosis and to choose the most appropriate management protocol, with a specific focus on on-farm diagnostic tools.

## 1. Introduction

Mastitis, the inflammation of the mammary gland, is one of the most frequent diseases affecting dairy cows worldwide. It is responsible for approximately 60–70% of all antimicrobials administered on dairy farms [1,2]. It leads to severe economic losses, both direct and indirect. Direct costs associated with the treatment of the disease do contribute to overall total losses. Nonetheless, indirect costs such as discarding contaminated milk, losses in future milk production and quality, and increasing culling risk are substantially more consequential [3]. Several studies tried to estimate the losses due to mastitis infection. Two main different approaches have been used: observational and modelling or simulation approach. Indeed, several factors affect the estimation of costs, such as differences in the farming system, herd management and implementation rules of treatments, and differences both in the upper limit of milk production and in milk pricing according to tank SCC and other differences in prices of production factors [4]. Worldwide, published estimates of the economic losses of clinical mastitis range from EUR 61 to EUR 97 per cow on a farm, with large differences between countries; for example, in the Netherlands, losses due to clinical and subclinical mastitis varied between EUR 17 and EUR 198 per cow per year [5]. Underestimating the economic losses of mastitis can be regarded as a general problem in the dairy sector. To improve the adoption rate of advice and lower the incidence of mastitis, it is important to show farmers the real economic losses of mastitis on their farm [6].

It also represents a public health concern, considering that approximately 62% of isolated mastitis-causing agents are resistant to at least one antimicrobial agent, and that some of them are zoonotic agents [7,8,9]. For this reason, SCC has been set as a key component of national and international regulation for milk quality and commercialization. According to the US Food and Drug Administration, SCC official threshold for internal consumption is 750,000 cells/mL [10]. The US Pasteurized Milk Ordinance mandates that milk samples from all farms are officially tested at least four times in each six-month period. Under the current system, if two of four monthly tests exceed the 750,000 cells/mL, the producer is notified by the state regulators, and if three of five tests exceed the limit, then their milk permit is revoked [11]. However, recently, a European Union Health Certification Program has been introduced, specifying an SCC threshold of 400,000 cells/mL for companies exporting products into the EU [12]. Within the EU, the eligibility to supply raw milk for processing of dairy products for human consumption is governed by Regulations 853/2004 [13] and 854/2004 [14] using measurements of raw bulk tank milk SCC (BTSCC) to determine eligibility. According to these, BTSCC must be lower than 400,000 cells/mL, stated as a rolling geometric average over two or three months, with at least one sample collected per month. Moreover, in some countries, producers receive reimbursement for delivering milk with a low somatic count due to its more desirable technical characteristics and a longer shelf life [15].

Mastitis can be classified, according to its etiology, into environmental and contagious or, according to symptoms, into clinical and subclinical [2].

Being a multifactorial disease, mastitis susceptibility is given by multiple factors such as age, parity, lactation stage, milk yield, and udder anatomical dispositions. Two of the main factors are immunological condition and reactivity of the mammary gland. Consequently, the clinical manifestations as well as its further course depend on the interplay between the innate resistance and adaptive immunity of the dairy cow and the type, concentration, and virulence of udder pathogens [16].

Major contagious mastitis pathogens include *Staphylococcus aureus*, *Streptococcus agalactiae*, and *Mycoplasma bovis*. Environmental mastitis pathogens include a wide range of organisms, such as coliforms (*Escherichia coli*, *Klebsiella*, *Enterobacter*, and *Citrobacter*), environmental streptococci (*Streptococcus uberis* and *Streptococcus dysgalactiae*), *Trueperella pyogenes*, non-*aureus* staphylococci (NAS), and others such as *Pseudomonas*, *Proteus*, *Serratia*, *Aerococcus*, *Listeria*, yeast, and *Prototheca* [17].

A 2019 study [18] showed how pathogens were associated with specific histopathological patterns. Lymphoplasmacytic and suppurative mastitis were the main patterns observed with involvement of *Streptococcus* spp., coagulase-negative staphylococci (CNS), *S. aureus*, *S. agalactiae*, *S. uberis*, and *Corynebacterium bovis*. The pyogranulomatous pattern presented different forms depending on the agent involved and was primarily associated with *S. aureus* and *Nocardia* spp. The cases of abscedative mastitis were predominantly caused by *T. pyogenes*. The necrosuppurative pattern was predominantly associated with environmental bacteria producing endotoxins, such as *E. coli*. Granulomatous mastitis had the lowest frequency of cases and was occasionally associated with *Mycobacterium* spp.

In healthy lactating cows, macrophages are the dominant cell type found in milk and mammary tissues [19]. When bacteria enter the mammary gland through the teat canal, they multiply in the milk, enhancing an inflammatory response. Bacterial toxins, enzymes, and cell-wall components may have a direct effect on the function of the mammary epithelium. They also stimulate the production of numerous mediators of inflammation by inflammatory cells that may be directly involved in the pathogenesis of the disease [20]. These mediators include prostaglandins, leukotrienes, serotonin, histamine, complement components, and cytokines, such as tumor necrosis factor, interleukin-I, interleukin-6, interleukin-8, granulocyte colony-stimulating factor, and granulocyte-macrophage colony-stimulating factor. These bioactive compounds have various activities locally in infected tissues that direct movement and localization of neutrophils (and other leukocytes) from blood to milk [21]. The neutrophil recruitment into lacteal secretion results in the increase in the somatic cell count (SCC) [22].

Clinical mastitis is characterized by local and systemic clinical signs, alteration in milk composition and appearance, and elevated SCC. Subclinical mastitis manifests through production losses, lowered milk quality, and high SCC [23]. Dairy cows with subclinical mastitis might be overlooked because infected cows may not present any clinical signs. Moreover, available data demonstrate that subclinical mastitis is 15 to 40 times more prevalent than clinical mastitis. Thus, it is fundamental to find an accurate, effective, fast, and economic diagnostic tools in order to detect mastitis and apply an appropriate management protocol also in the absence of clinical signs [24].

The objective of this review is to describe the current approach to control, detect and characterize mastitis in the dairy cow, focusing on emerging diagnostic and on-farm tools, in order to promptly identify intramammary infection and to choose the most appropriate management protocol, also considering the increasingly concern for antimicrobial resistance.

## 2. Diagnosis and Control

### 2.1. Diagnosis

An efficient and effective mastitis control program requires the early detection of infection. This can be obtained by understanding the pathogenesis, discovering new sensitive tests for early screening, and adopting good managemental practices to reduce the chance of transmission of the infection from sick to healthy quarters [25].

Early detection of mastitis and identification of the causative agent are crucial for control and treatment [26]. These measures are fundamental in reducing costs, reducing losses in milk yield and milk quality, and increasing the cure rate of the infected animals [27]. Several studies demonstrated that a consequent early treatment significantly limited the severity of the disease and, in many cases, prevented the appearance of any visible signs of infection [28].

SCC and bacteria detection are the main tools to diagnose mastitis. In particular, bacteriological culture and PCR analysis are considered the gold standard.

Mammary infection is determined with a cell number above 200,000 cells/mL [29]. In addition, long-term elevated SCC levels suggest that the affected quarters are in a state of chronic inflammation and may interfere with the development of lactating tissues [30]. SCC varies during lactation, reaching a nadir at 30 DIM, coinciding with the peak of lactation, and gradually increasing in the remaining period [31]. Some authors have attributed high cell counts early in lactation to excessive shedding of epithelial cells in a small volume of milk. In fact, the mammary gland resumes function after a dormant period and in late lactation to a mere concentration of cells in a smaller volume of milk as milk production declines [32]. SCC has been proved to also be affected by parity, milking time, milking frequency, season, and udder infection [33,34]. A study has evaluated the variation in SCC patterns relating to the specific pathogen. In accordance with previous studies, for example, the authors found that intramammary infections (IMI) with *S. aureus* and *S. uberis* were often characterized by a long duration and high SCC; in the case of *E. coli* mastitis, instead, there was an association with short peaks [35].

New technologies, such as metabolomics, are emerging as tools for diagnosing and preventing mastitis. Metabolomics evaluates different changes in metabolites in affected cows in order to find effective biomarkers for timely and accurate prevention. Some interesting findings have been highlighted in two studies. The first has applied gas chromatography–mass spectrometry (GC-MS) on blood, finding as reliable the subclinical mastitis markers valine, serine, tyrosine, and phenylalanine 4–8 weeks pre-partum and valine, isoleucine, serine, and proline 4–8 weeks post-partum [36]. The second study has applied Nuclear Magnetic Resonance Metabolomics (NMR) on milk, finding that lactate, acetate, BHBA, butyrate, and isoleucine were in a higher concentration in high-SCC samples. On the other hand, for the same samples, lactose, hippurate, and fumarate were at lower levels than in milk with low SCC levels [37].

Also, recent molecular techniques such as MALDI-TOF and commercially available quantitative real time PCR (rtPCR) tests are becoming more common. MALDI-TOF is used to identify isolated organisms to genus and/or species level with a lower incidence of misclassification errors compared to traditional phenotypic speciation methods for some mastitis pathogens. Commercially available rtPCR tests reveal the advantages of being directly applied to milk samples containing preservatives: this procedure not only facilitates storage and shipment at room temperature, but it also enables the identification of organisms that are difficult to culture in a timely fashion [38].

Another group of widely investigated potential biomarkers are Acute Phase Proteins (APPs), already commonly employed as clinical biomarkers of inflammation in serum, such as milk isoforms of serum amyloid A (M-SAA) and haptoglobin (HP). Other proteins indicated as suitable mastitis markers are lactoferrin (LF) and cathelicidins (CATH) [25,39]. Some enzymes, such as Lactate dehydrogenase (LDH) and N-Acetyl-β-d-Glucosaminidase (NAGase), also seems to be reliable indicators of mastitis [40].

### 2.2. Control

The control program must include the strategic use of antimicrobials to curtail the problems of antibiotic residue in milk and antimicrobial resistance [41]. This implies the implementation of preventive measures. Most prevention activities focus on milking time and procedures. The use of management practices which reduce bacterial contamination of teat ends is a basic aspect of mastitis control. Pre-milking sanitation had usually been performed by washing udders, teats, pre-dipping, and post-dipping with water or disinfectants. Post-milking teat antiseptic was regarded as the single most effective practice to control IMI of lactating cows. This includes dipping or spraying the teat with an antiseptic formulation after milking. Antiseptic is sometimes associated with a filming agent in order to physically isolate the teat from the environment [23,42]. Furthermore, nutrition is an important factor in the resistance against disease. Deficiency of some trace substances and vitamins such as selenium, copper, zinc, and vitamin E have been found to be predisposing factors for mastitis. Dietary recommendation and possible supplementation of the lacking substances might significantly improve udder health [43]. Vaccination represents another preventive strategy. The use of vaccination, particularly with autogenous killed whole-cell vaccines, to try to control mastitis in dairy cattle is common. Several efforts have been made to develop a vaccine against mastitis, but few have claimed satisfactory outcomes, either in the field or on backyard farms. The main problem is that a single vaccine will not prevent mastitis caused by the multiple pathogens with different mechanisms of pathogenesis; if immunity is good against a certain agent, it does not protect from another one [44].

Nonetheless, the main strategy to treat mastitis still remains the administration of antibiotics. These can be given either by intramammary infusion or by intramuscular or intravenous injection [45]. For successful antibacterial mastitis therapy, the active medicine must achieve and conserve concentrations above the minimum inhibitory concentration (MIC) at the center of infection for long enough to breakdown the production and toxin-producing phase of the causal pathogen. This may be prohibited by several factors that include pathological changes in the udder parenchyma and mastitogenic bacterial and related factors [44,46]. Antibiotic administration can be performed either during lactation or the dry period. Treatment may be given during lactation for clinical mastitis and for some specific cases of subclinical mastitis. Treatment of clinical mastitis usually involves administering antibiotics to eliminate the causative organism from the quarter. A clinical response must be perceptible within 5–7 days [47]. Antibiotic therapy at drying-off has a double interest: first, to treat IMI in quarters contaminated at drying off (curative effect), and second, to prevent new infections at the beginning of the dry period [48,49]. For many years, blanket dry cow therapy (BDCT) has been the most common approach to mastitis control. This implied the infusion of all quarters of all cows with a long-acting intramammary antimicrobial at the end of the lactation. As concerns regarding the emergence of antimicrobial-resistant pathogens are on the rise, a protocol known as selective dry cow therapy (SDCT) has been applied. This establishes the treatment only of quarters with IMI at dry off [50]. Since SDCT was introduced, different methods of selecting infected cows or quarters have been reported: bacteriological culture in the laboratory, somatic cell counts (SCC), on-farm diagnostic tools and/or history of clinical mastitis. Many studies have demonstrated that the use of SDCT does not increase the risk of IMI at calving if internal teat sealants are used for all cows [51]. SDCT thus represents an effective treatment approach to reduce antimicrobial usage without affecting animal welfare or the farm economy [52].

## 3. On-Farm Diagnostic Tools

### 3.1. Mammary Ultrasound

Ultrasonography is a non-invasive method for the diagnostics of various physiological and pathological conditions of the reproductive organs of ruminants [53]. It has also more recently been applied to the measurement of subcutaneous fat and for the diagnosis of respiratory diseases [54,55].

Many studies have demonstrated the utility of mammary ultrasound in identifying the presence of lesions or alterations of the mammary parenchyma and the teat, as well as in providing a valid indication of the outcome of treatment based on this evidence. Moreover, the possibility of pre-identifying animals for which intramammary antibiotic treatment could be ineffective would be an important element both in terms of antimicrobial resistance and for the economic impact of the disease itself [56,57,58,59]. The echographic examination of the mammary gland parenchyma in ruminant animals is performed primarily via the direct contact technique (transcutaneous echography). For this purpose, a linear, convex, or sector transducer can be used [56]. Ultrasonography of the normal udder parenchyma shows homogenous hypoechogenic parenchyma with interspersed anechoic blood vessel, milk alveoli, and lactiferous duct (Figure 1).

The gland cisterns appear as a large homogenous anechoic area with few hypoechoic dots, which correspond to the milk [60]. A study performed on 52 dairy cattle proposed the classification of the parenchyma alterations in seven groups: physiological sonographic pattern (medium homogeneous echogenicity, isolated anechoic lactiferous ducts and vessels), non-homogeneous hyperechoic and few lactiferous ducts, abscesses (round, well-defined structures of varying size with a distinct capsule and hypoechoic content), non-homogeneous with hyperechoic zones and gas formation, hematoma (large septal spaces filled with anechoic up to hypoechoic fluids), non-homogeneous hypoechoic areas, and spotted hypoechoic areas. Consequently, a grading of the sonographic changes was also proposed as follows: physiological sonographic pattern, mild changes (only small percentages of the udder tissue are affected, with few obvious sonographic changes), moderate changes (larger areas of the udder tissue affected with more pronounced sonographic changes) and severe changes (normal udder structure is hardly extant anymore) [59]. Another study tried to examine whether ultrasonography could be used to predict the outcome of clinical mastitis. Udders were examined and classified according to on one or more of the following three signs: a hyperechoic spot in the parenchyma area, structural changes in the milk ducts, and non-homogenous parenchyma. According to the statistical analysis, only the presence of “non-homogeneous parenchyma” could be selected as a factor that predicts a marketable milk production [57].

An example of quarter with *E. coli* infection can be seen in Figure 2, which shows the presence of hyperechoic spots and irregular aspect of the parenchyma.

Concerning B-mode echography of the teat in ruminants, instead, this is most often performed using the water bath technique and vertical scanning [56]. Under physiological conditions, the teat wall shows three layers; the skin appears as a thin, hyperechoic line, followed by the muscular/connective tissue layer containing blood vessels showing a thicker, homogenous, hypoechoic layer, and the innermost mucous membrane appears as a hyperechoic layer. A study evaluated the alterations of the teat structures in the case of subclinical mastitis. The teat cistern appeared as an anechoic area with few hypoechogenic dots according to the presence of milk content. There was an irregular outline of the teat canal and cistern, homogenous hypoechogenic contents, and loss of the three-layered appearance of the affected teat wall [60].

### 3.2. Blood Gas Analysis

The clinical status of the animal is equally crucial in assessing the success rate of treatment. Hematobiochemical testing is a useful and reliable indicator of the severity of syndromic conditions and is also a valuable tool in the diagnosis and prognosis of many diseases. Several studies have focused on the evaluation of the main alterations during mastitis, with the intention of also extending the use of blood parameters to this disease. The application of an Emogas analyzer would provide the possibility of obtaining prognostic index also in field condition and in real time [36,61,62,63,64]. Blood gas analysis is already extensively used in the evaluation and management of multiple pathologic condition. For example, in calves, it is useful to assess the acid–base status and to establish an appropriate treatment protocol in cases of respiratory disease and diarrhea [65,66,67].

Very recently, researchers investigated the variation of blood parameters at different milk production. Interestingly they found a direct correlation between pO_2_ and milk production. This can probably be justified because the ability to transport oxygen is increased in cattle producing higher amounts of milk. Also, HCO_3_^−^ was higher in high-producing cows. Finally, high glucose blood concentration was associated with low milk production, probably induced by the high rates of utilization needed to fulfill the requirements of the mammary gland for higher milk synthesis [68].

Several authors have investigated the application of ultrasound mammary evaluation in the case of mastitis. A study of 2022 tried to use the blood gas analyzer in cattle with high SCC to evaluate permeability in the mammary gland. The authors analyzed sodium (Na) and potassium (K) in particular, as they are considered to be important indexes for mammary permeability evaluation. The results showed higher levels of K, and a higher ratio of Na to K in animals with high SCC, suggesting the use of blood gas parameters for a rapid evaluation of the mammary gland health status [62]. Additionally, another article of 2016 showed levels of calcium, phosphorous, and zinc were significantly lower in the blood serum of infected cattle than in healthy cattle [69].

Finally, a study highlighted a decrease in glucose blood concentration in the case of clinical mastitis. Several authors explain this alteration with the concurrent presence of negative energy balance (NEB), very frequent in the transition period. Such NEB leads to significantly reduced serum glucose levels, thus affecting the energy supply of the immune system and predisposing animals also to mastitis onset [70,71]. Few other hypotheses have been made. For example, another possible explanation is that blood flows to the infected udder and result in a limited supply of blood sugar secreted by the cell. Another hypothesis is that glucose is transported from the milk to the extracellular pathway to maintain the osmotic balance between the extracellular environment and the milk with the increase in Na and chlorine (Cl) when the body suffers from mastitis. In addition, hypoglycemia may be due to the accumulation of cells at the site of infection, which depletes the glucose in the local area [36].

### 3.3. Electrical Conductivity

Electrical conductivity (EC) is a measure of the resistance of a particular material to an electric current. The electrical resistance of an electrolyte solution is defined as the resistance of a cube of the solution 1 cm^3^ in volume. The conductivity is the reciprocal of the resistance. Resistance (impedance) is measured in ohms and is calculated by dividing voltage by amperes. Conductivity is measured in Siemens and is calculated by dividing ampere by voltage [72]. In milk, it is determined by the concentration of anions and cations, with Na, K, and Cl being the most important [73].

When a cow is exposed to IMI, tight junctions and the active Na, Cl, and K pumping system, characteristic of the mammary gland, are destroyed, leading to a leakage of these ions into the milk [74].

Several studies have demonstrated how EC can be influenced by several factors, such as milk fat content and other pathological processes. In particular, a 2009 study highlighted how milk fever was associated with the greatest increase in EC and the longest for cows diagnosed with pneumonia. Other metabolic and digestive problems such as ketosis, off feed, left displaced abomasum, as well as retained placenta, lameness, and other diseases were also associated with a significant increase in EC. Nonetheless, results also demonstrated a significant increase in EC as early as three days before detection of mild mastitis with other methodologies [75].

Different methods are applied to evaluate EC. In recent studies, an individual conductivity > 6.5 mS/cm and a difference between quarters with the highest and lowest conductivity > 1.0 mS/cm have been considered the cut-off for infection diagnosis [76].

A study highlighted that the accuracy of electrical conductivity compared favorably with other indirect methods, especially considering stripping milk rather than foremilk obtained after milk ejection [77]. Another study compared EC in case of mastitis caused by primary and secondary pathogens, observing in both significant differences with uninfected quarters. The authors also evaluated differential and absolute conductivity; the results from absolute conductivity were generally similar and sometimes superior to those from differential conductivity, especially when stripping were used [78]. Additionally, a research study from 1998 highlighted differences in infections detection relating to the bacteria. In fact, *S. aureus* and *S. uberis* were more readily detected than CNS infections. This may arise from less epithelial damage and inflammation generated by CNS and/or the possibility that such infections were localized in the teat canal, teat sinus or gland cistern and ductal regions [79].

The ability of EC to detect subclinical mastitis is still controversial, and further studies are needed, even if the association with other parameters significantly improve predictability [74,80].

### 3.4. California Mastitis Test

The California Mastitis Test (CMT) is an elementary, low-cost, quick evaluation that can be conducted on farm and is commonly used for identifying subclinical mastitis during milking [81]. Milk is appropriately collected in the CMT paddle, and an equivalent amount of the reagent (sodium alkyl aryl sulfonate) is added using a horizontal swirling motion for about 10–15 s. Sodium alkyl aryl sulfonate is an anionic surfactant that decreases surface tension, changes the structure and conductivity of cell membrane and nucleus, interferes with the osmotic balance, blocks oxidization, stimulates proteolytic enzymes, and increases milk viscosity [82]. The formation of gel reflects a leucocyte count of 200,000 to 5,000,000, indicative of IMI. Moreover, the purple color that results from the test is generally more intense in samples from infected quarters. This is because such samples have a basic or alkaline pH. The deep purple color indicates abnormal composition of milk due to mammary gland infection [83]. A study of 2014 also demonstrated a significant correlation between SCC and CMT, in which higher CMT scores corresponded to elevated trends of SCC [84]. These findings are supported by another article of 2012. Researchers divided CMT reaction into three groups, from the weakest to the strongest reaction (CMT+, CMT++, CMT+++) and compared them with SCC. In accordance with the previous article, they highlighted a direct correlation between these two diagnostic methods [85]. The use of the CMT to identify infected quarters has been extensively validated in cows that were not in early lactation [86]. Moreover, the possibility to identify not only clinical, but also subclinical mastitis at the end of lactation, before the dry off, would be very important in order to perform SDCT. A 1981 study reported that positive CMT scores in milk samples obtained several weeks before drying off predicted accurately IMI in over 80% of cases [87]. Another study demonstrated a high negative predictive value for IMI associated with the major pathogens performing CMT on the day of dry off [88]. Similar results have been highlighted in a 2012 article, in which CMT presented good negative predictive values, good sensitivity, even if positive predictive values and specificity were low [89]. Duplicate or multiple milk samples might positively affect the outcome of the test.

### 3.5. On-Farm Culture

On-farm culture (OFC) is a system which allows veterinarians to address the producers to the best strategic treatment decisions for clinical mastitis cases, without delay between the submission of milk samples and the reporting of results implied by laboratory culture. Initial on-farm culture systems were based on blood and MacConkey agar plates, which facilitated the categorization of microorganisms into Gram-positive, Gram-negative, or no growth within 24 to 32 h [90]. Several studies demonstrated its usefulness in guiding the decision of the most appropriate treatment decreasing intramammary antibiotic use, both in the short and long term. A study showed a decrease in intramammary antimicrobial use by half and a tendency to decrease withholding time by 1 day. This did not significantly affected days to clinical cure, bacteriological cure risk, and new infection risk or treatment failure risk within 21 days after the clinical mastitis event [91]. A companion paper, focused on long-term outcomes, demonstrated that selective treatment of clinical mastitis based on on-farm culture resulted in no difference, for example, concerning the recurrence of clinical mastitis in the same quarter, SCC, milk production, and cow survival for the rest of the lactation after clinical mastitis [92].

Chromogenic culture media are alternatives to rapid microbiological identification, as they make it possible to presumptively differentiate bacterial species and/or groups according to colony color [93]. The chromogenic substrate, when it contacts a specific microorganism after undergoing hydrolysis, releases a dye that sets in the microbial colonies, differentiating them by color [94]. According to a study, chromogenic media presented 100% Sensibility and 99.8% Specificity for the identification of *S. aureus* in clinical mastitis milk samples [95]. A study conducted on cases of subclinical mastitis caused by Gram-positive bacteria, showed satisfactory diagnostic performance results for the identification of the main pathogens, despite the limitation for the identification of *S. aureus* [96]. Another study performed comparing bi- and tri-plate both on clinical and subclinical mastitis suggested adequate accuracy for identification of *S. agalactiae*, *S. dysgalactiae*, *S. uberis*, *E. coli*, *Klebsiella* spp., and *Enterobacter* spp. [94]. However, the type of microorganism causing bovine mastitis and mastitis presentation influence diagnostic accuracy. Also, experience in milk microbiology substantially improved the interpretation of on-farm culture results, indicating that the observers’ experience is crucial to facilitate appropriate management decisions when adopting a selective treatment protocol [96]. Therefore, the use of chromogenic culture media on the dairy farm routine could be an alternative for the rapid identification of subclinical mastitis.

### 3.6. Infra-Red Thermography

The physical principal at the basis of the Infra-Red Thermography (IRT) is that a body that has a temperature higher than absolute zero emits electromagnetic radiation in the infrared spectrum. The relationship between the energy emitted by the body surface, the wavelength of this radiation and the temperature of the body is mathematically described. This radiation can be detected through a sensor array and used to build a thermographic image where the intensity, or the color, of each pixel is proportional to the corresponding temperature of the surface observed [97,98]. In a thermogram, the warmest region appears as white or red, whereas the coolest region appears as blue or black [99].

In bovine medicine, IRT is already extensively used for heat detection and prediction of ovulation in cows, detection and assessment of lameness, assessment of animal welfare, and feed utilization efficiency [100].

Mastitis is an inflammatory process of the mammary gland, leading to the production of prostaglandins and, consequently, to the increase in body and udder surface temperature, which can be detected via IRT (Figure 3) [101].

Recently, several studies have also established a significant positive correlation between SCC and udder temperature. Performing thermal imaging, it is important to consider that IRT images can be influenced by several factors, such as mechanical brushing of the udder, direct solar radiation and wind speed, parity, stage of lactation, and pregnancy. It has also been shown that the temperature of the thermal images of the forequarters region of the udder had a greater correlation with SCC. This is because the rear region of the udder is more exposed to climatic variables and physical damage during milking, which may overestimate udder temperature during the IRT analysis. For this reason, it is fundamental to calibrate the thermal imager and to establish a standardized distance and acquisition method [101,102]. Commonly, an emissivity of 0.98 throughout is employed in accordance with previously published studies carried out on cow udders [97]. For the acquisition of thermogram, a study of the 2022 suggested photographing the udder from three angles to identify all quarters, and to photograph each one at least three times to exclude a low-quality image. Each angle had a range of distance and a range of tilt angle. Depending on the structure of the udder and the position of the cow in the stall, the specialist placed the thermal imager at 30–100 cm from the udder and teats of the animal at an angle of −15° to +15°. The choice of distance and angle of inclination of the thermal imager was based on the viewing angle of the thermal imager, the physiological dimensions of the udder, and the availability of all udder teats in the image. Thermographic images were obtained before the start of milking and after the end of milking, even if no significant difference between temperature of left and right udder quarters before and after milking was found [100,102].

## 4. Conclusions and Future Perspectives

Nowadays, mastitis still represents one of the most frequent diseases of dairy cows, having well-recognized detrimental effects on animal wellbeing and farming economy, and being one of the main causes of antimicrobial use in dairy farms. Due to the growing concern for antimicrobial resistance, regulations have been implemented nationally and internationally to reduce unnecessary antibiotic use. Not all IMIs require antibiotic treatment, and their administration must be based on the culture and sensitivity results rather than empirical therapy. Moreover, therapy success depends on several factors such as causal agent, parity, stage of lactation, other systemic diseases, and mammary parenchyma alterations. The management of mastitis must be under constant and continuous control of the veterinarian. This could certainly be facilitated by the possibility of establishing which animals are prone to have an improvement in treatment success; an improved treatment success would also result in a reduction in antimicrobial administration. Some on-farm diagnostic tools are already being routinely used to apply SDCT, such as CMT and OFC. IRT is a noninvasive and rapid method and can be of great value in early mastitis detection for optimum response from treatment. Even if it is an indirect measure, it has relatively good sensitivity, specificity, and positive and negative predictive values. Additionally, it can perceive changes in skin surface temperature in response to varying degrees of severity of the mammary gland infection. Ultrasound mammary evaluation has proved to be effective in identifying the presence of lesions or alterations of the mammary parenchyma and the teat. Such alterations might affect the prognosis of the animal. Also, blood gas analysis would provide interesting parameters to evaluate on farm and in real time the clinical conditions of the affected cows. These diagnostic techniques are still emerging in this field, and their application to detect mastitis and establish whether to treat a case of mastitis or not certainly has to be improved. Further studies are needed in order to establish objective parameters of detection and prognostic index.

## Figures and Tables

**Figure 1 animals-13-02538-f001:**
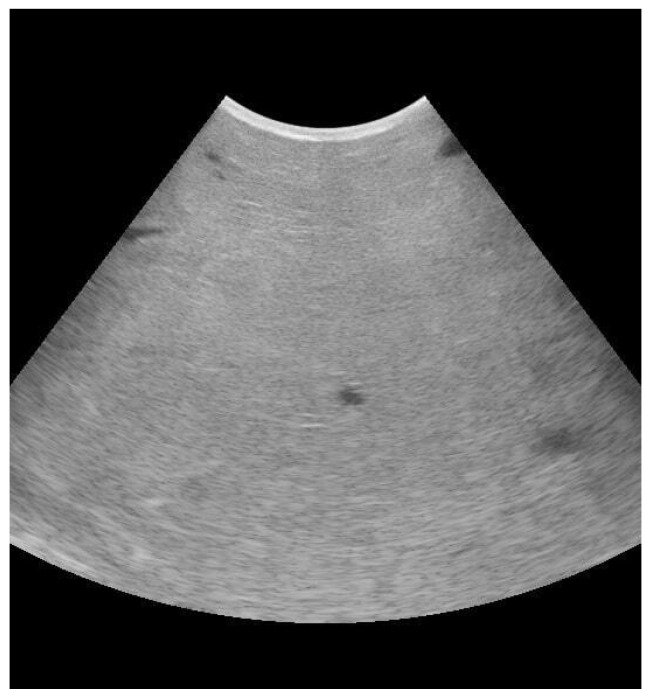
Mammary ultrasound evaluation with longitudinal scanning (B mode). Frequency: 6.5 MHz. Depth: 150 mm. Focus: 50 mm.

**Figure 2 animals-13-02538-f002:**
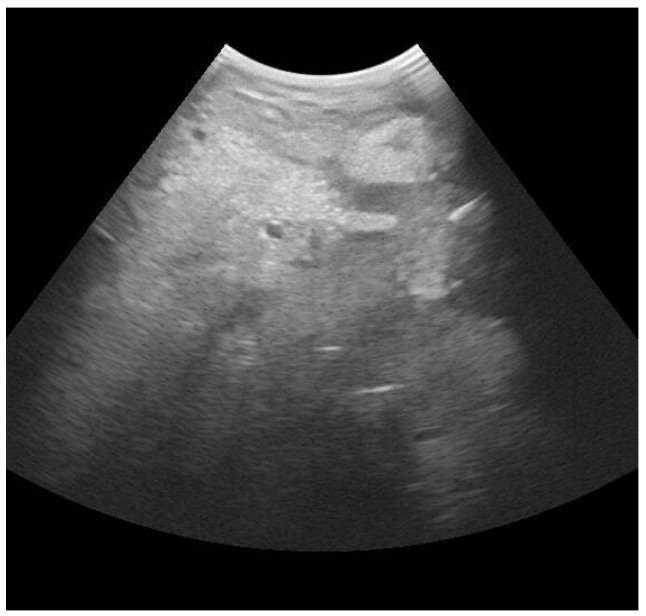
Mammary ultrasound evaluation with longitudinal scanning (B mode) of a quarter with *E.coli* infection. Frequency: 6.5 MHz. Depth: 150 mm. Focus: 50 mm. Presence of multiple hyperechoic spots and irregular parenchyma.

**Figure 3 animals-13-02538-f003:**
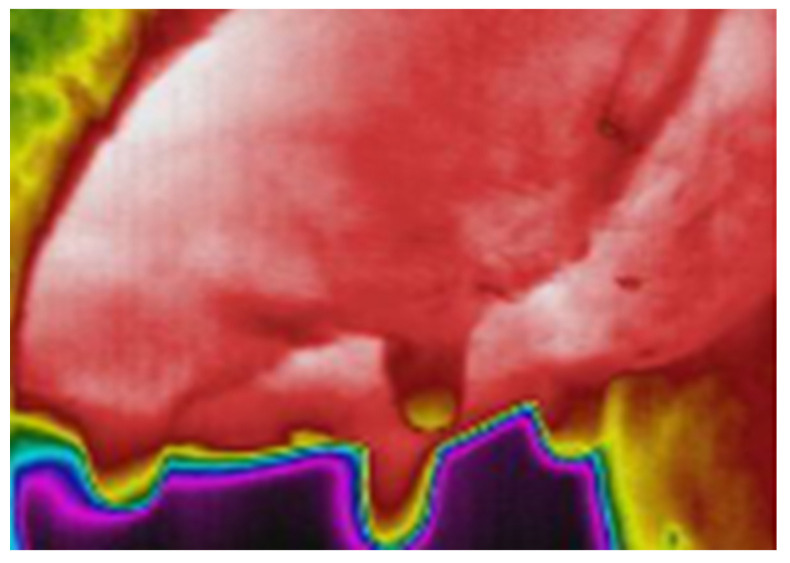
Infra-Red Thermography of affected quarter. The presence of red color in correspondence with the teats and orifices indicates increased surface temperature. No relevant difference between udder and teat temperature is detected, which is present in healthy quarters.

## Data Availability

Not applicable.

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
