# Peer review of "Mastitis in Dairy Cattle: On-Farm Diagnostics and Future Perspectives"

_animals, 2023, doi:10.3390/ani13152538_

Round 1
Reviewer 1 Report
Dear authors,
The paper represent one of the main dairy farm problems, so, this review is very important to know some innovative diagnostic tools, however, I suggest emphasizing or strengthening in your discussion the importance of mastitis prevention measures and early diagnosis of sick animals. The CMT has extensive benefits despite the years.
There are some comments in the text of atachment file

Author Response
In accordance with your suggestions, we modified the article as follows:
- Dear authors, the paper represents one of the main dairy farm problems, so, this review is very important to know some innovative diagnostic tools, however, I suggest emphasizing or strengthening in your discussion the importance of mastitis prevention measures and early diagnosis of sick animals. The CMT has extensive benefits despite the years.
- AU: Thank you, very much for you comment, the interest you declare for the topic is important and pleasing for us. To emphasize the importance of prevention and timely diagnosis we improved the manuscript with new paragraphs (L131-136 and L189-200) according to these references:
-Pyörälä, S. New Strategies to Prevent Mastitis. Reproduction in Domestic Animals 2002, 37, 211–216, doi:10.1046/J.1439-0531.2002.00378.X.
-Tiwari, J.; Babra, C.; Kumar Tiwari, H.; Williams, V.; De Wet, S.; Gibson, J.; Paxman, A.; Morgan, E.; Sunagar, R.; Isloor, S.; et al. Bovine Mastitis Preventive and Therapeutic Strategies. J Vaccines Vaccin 2013, doi:10.4172/2157-7560.1000176.
- L 34 Add another different keywords to amplify potential citation
- AU: Thank you for the suggestion, “mammary ultrasound, blood gas analysis, Infra-Red Thermography, California Mastitis Test, electrical conductivity” have been added as new key word. (L 35-36)
- L 48 Italicize Streptococcus agalactiae and Mycoplasma bovis
- AU: Thank you, terms have been italicized (L84-85)
- L63 Describe with more details what is order of lactation, its stage and anatomical dispositions
- AU: Thank you, we tried to specify better the concepts “Such as age, parity, lactation stage, milk yield, udder anatomical dispositions” (L78-79)
- L107 Change by mL
- AU: Done, thank you (L139)
- L110 Justify the SCC nadir at 30 DIM
- AU: Thank you for the comment. We justified the statement according to “Kennedy, B.W.; Sethar, M.S.; Tong, A.K.W.; Moxley, J.E.; Downey, B.R. Environmental Factors Influencing Test-Day Somatic Cell Counts in Holsteins. J Dairy Sci 1982, 65, 275–280, doi:10.3168/JDS.S0022-0302(82)82188-7.” as written in L142-147 here reported:
“reaching a nadir at 30 DIM, coinciding with peak of lactation, and gradually increasing in the remaining period. Some authors have attributed high cell counts early in lactation to excessive shedding of epithelial cells in a small volume of milk. In fact, the mammary gland resumes function after a dormant period and in late lactation to a mere concentration of cells in a smaller volume of milk as milk production declines.”
- L260 Change in three days
- AU: Done, thank you (L339)
- L283 and following. Check the reaction lecture time
- AU: Thank you, the correct lecture time is 10-15 seconds. (L364)
- L300 Reconsider the role of “producer” as factor to increase the risk of antimicrobial resistance. The participation of a Veterinarian is so important to management the herd health.
- AU: Thank you for the comments. The role of the veterinarians is fundamental in the management of dairy herds, representing the guide for producers in the choice of the most appropriate treatment. So, we rephrase the text as follow: “On-farm culture (OFC) is a system which allows veterinarian to address the producers to the best strategic treatment decisions for clinical mastitis cases.” (L388-389)
- L365 Change in three
- AU: Done, thank you (L454)
- L 385 and following. It is very important to mention the activity of a Veterinarian daily participation as dairy health responsible.
-
- AU: Thank you for the comment. To be able to know the comprehensive situation of the farm and to manage in the most suitable way the dairy herd, the activity of the veterinarian has to be as accurate as possible. So, we implemented the paragraph as follow: “The management of mastitis must be under constant and continuous control of the veterinarian. This could certainly be facilitated by the possibility of establishing which animals are prone to have an improvement in treatment success; an improved treatment success would also result in a reduction of antimicrobial administration.” (L474-477)
Reviewer 2 Report
Mastitis in dairy cattle: on-farm diagnostics and future perspectives.
Review paper on diagnostic methods for mastitis, both clinical and subclinical, with special emphasis on reducing the use of antibiotics. The early diagnosis of sick animals or quarters will only allow them to be treated, and even if possible, during the drying period, thus avoiding public health problems derived from the indiscriminate use of antibiotics.
Although it does not go into depth in any of the chapters, the review that has been done can give a general idea of the diagnostic systems for mastitis in dairy cows.
Maybe some parts should be rewritten again, avoiding repetitions, and giving a more global idea. Sometimes experiences from different authors have been added, but interrelated and without correct wording.
I miss amplifying the economic losses derived from cows affected with mastitis. I would recommend referring to the SCC limits, both in the European Union and in the USA, for the commercialization of milk (Regulation (EC) nº 853/2004 of the European Parliament and of the Council of 29 April 2004 laying down specific hygiene rules for food of animal origin). And perhaps also add a publication where he expresses more broadly the relationship between the RCS and the CMT.
Other indications.
L-16. “Selective dry cow therapy reserves antimicrobial treatment for cows or quarters suspected of having an intramammary infection, while uninfected cows or quarters do not receive antimicrobial treatment”. Change for: “Selective dry therapy reserves antimicrobial treatment for cows or quarters suspected of having an intramammary infection, applied after the last milking, while uninfected cows or quarters do not receive antibiotherapy”.
L-37 and following: Reorder the Introduction.
– Mastitis, the inflammation of the mammary gland…
– Mastitis can be classified based on etiology into
– Mastitis is a multifactorial disease, and susceptibility is…
– Major contagious mastitis pathogens include Staphylococcus aureus…
– A study of 2019 showed how pathogens… (Maybe delete. It contributes nothing to the objective of the review). Add the reference here, at the beginning of the sentence.
– In healthy lactating cows, macrophages are the dominant cell type…
– Clinical mastitis is characterized by local and systemic clinical signs…
L-55. Define what is CNS the first time you cite.
L-277. Change for: “The ability of EC to detect subclinical mastitis is still controversial and further studies are needed, even if the association with other parameters can significantly improve predictability.”
L-290. “Moreover, the possibility to identify subclinical mastitis at the dry off would be very important in order to perform SDCT.”
I don't understand. How can clinical mastitis be diagnosed on drying by CMT? I recommend rewriting again.
L-332. Define what is SCM the first time you cite.
L-281 and following: Add that the CMT test, in addition to the surfactant detergent, has a pH indicator, which can modify the color of the milk sample.
L-342 and following: “As affirmed before, mastitis is an inflammatory process of the mammary gland, resulting…” The explanation is strange. It could be made easier by saying that it is an inflammatory process, that there is an increase in temperature and that it is detected by the device.
L-350. “In bovine medicine, IRT is already extensively used for heat detection and prediction 350 of ovulation in cows, detection and assessment of lameness, assessment of animal welfare, 351 and feed utilization efficiency”. I recommend changing this paragraph of place, and placing it before the previous paragraph.
Author Response
In accordance with your suggestions, we modified the article as follows:
- Review paper on diagnostic methods for mastitis, both clinical and subclinical, with special emphasis on reducing the use of antibiotics. The early diagnosis of sick animals or quarters will only allow them to be treated, and even if possible, during the drying period, thus avoiding public health problems derived from the indiscriminate use of antibiotics. Although it does not go into depth in any of the chapters, the review that has been done can give a general idea of the diagnostic systems for mastitis in dairy cows. Maybe some parts should be rewritten again, avoiding repetitions, and giving a more global idea. Sometimes experiences from different authors have been added, but interrelated and without correct wording. I miss amplifying the economic losses derived from cows affected with mastitis. I would recommend referring to the SCC limits, both in the European Union and in the USA, for the commercialization of milk (Regulation (EC) nº 853/2004 of the European Parliament and of the Council of 29 April 2004 laying down specific hygiene rules for food of animal origin). And perhaps also add a publication where he expresses more broadly the relationship between the RCS and the CMT
-
- AU: Thank you for the comment, all the article has been checked. We tried to modify the manuscript to make it as clear as possible, and to correlate better different scientific studies.
Economic losses represent a fundamental aspect of this disease, representing one of the most affecting pathologies in the farm costs. We detailed this topic better at L45-56 according to these references:
-Seegers, H.; Fourichon, C.; Beaudeau, F. Production Effects Related to Mastitis and Mastitis Economics in Dairy Cattle Herds. Vet Res 2003, 34, 475–491, doi:10.1051/VETRES:2003027.
-Hogeveen, H.; Huijps, K.; Lam, T.J.G.M. Economic Aspects of Mastitis: New Developments. http://dx.doi.org/10.1080/00480169.2011.547165 2011, 59, 16–23, doi:10.1080/00480169.2011.547165.
-Huijps, K.; Lam, T.J.; Hogeveen, H. Costs of Mastitis : Facts and Perception. Journal of Dairy Research 2008, 75, 113–120, doi:10.1017/S0022029907002932.
SCC limit represents also a very important topic for the regulation of food commercialization. According to the country, different cut-off has been established, both for internal consume and exporting. We analyzed better this topic at L59-75 according to these references:
-U.S. Code of Federal Regulations, Title 21, Part 54, Title 21-Food and Drugs, Chapter I-Food and Drug Administration, Department of Health and Human Services, Subchapter B-Food for Human Consumption;
-Ruegg, P. Understanding the Changes in Bulk Tank Somatic Cell Count Monitoring;
-More, S.J.; Clegg, T.A.; Lynch, P.J.; O’Grady, L. The Effect of Somatic Cell Count Data Adjustment and Interpretation, as Outlined in European Union Legislation, on Herd Eligibility to Supply Raw Milk for Processing of Dairy Products. J Dairy Sci 2013, 96, 3671–3681, doi:10.3168/jds.2012-6182.
-REGOLAMENTO (CE) N. 853/2004 DEL PARLAMENTO EUROPEO DEL CONSIGLIO 29 Aprile 2004 Stabilisce Norme Specifiche in Materia Di Igiene per gli Alimenti Di Origine Animale; 2004;
-REGOLAMENTO (CE) N. 854/2004 DEL PARLAMENTO EUROPEO E DEL CONSIGLIO 29 Aprile 2004 Stabilisce Norme Specifiche per l’organizzazione Di Controlli Ufficiali Sui Prodotti Di Origine Animale al Consumo Umano; 2004;
-Neculai-Valeanu, A.S.; Ariton, A.M. Udder Health Monitoring for Prevention of Bovine Mastitis and Improvement of Milk Quality. Bioengineering 2022, 9.”
The relationship between RCS and CMT is very interesting, thank you for the suggestion. According to the authors written belove we implemented the paragraph at L371-376
“-Hoque, M.N.; Das, Z.C.; Talukder, A.K.; Alam, M.S.; Rahman, A.N.M.A. Different Screening Tests and Milk Somatic Cell Count for the Prevalence of Subclinical Bovine Mastitis in Bangladesh. Trop Anim Health Prod 2015, 47, 79–86, doi:10.1007/S11250-014-0688-0/TABLES/4.
-Kaşikçi, G.; Çeti̇n, Ö.; Barış BİNGÖL, E.; Can GÜNDÜZ, M. Relations between Electrical Conductivity, Somatic Cell Count, California Mastitis Test and Some Quality Parameters in the Diagnosis of Subclinical Mastitis in Dairy Cows. Turk J Vet Anim Sci 2012, 36, doi:10.3906/vet-1103-4. “
- L-16. “Selective dry cow therapy reserves antimicrobial treatment for cows or quarters suspected of having an intramammary infection, while uninfected cows or quarters do not receive antimicrobial treatment”. Change for: “Selective dry therapy reserves antimicrobial treatment for cows or quarters suspected of having an intramammary infection, applied after the last milking, while uninfected cows or quarters do not receive antibiotherapy”.
- AU: Thank you, we have changed for “Selective dry therapy reserves antimicrobial treatment for cows or quarters suspected of having an intramammary infection. The treatment is administered after the last milking, while uninfected cows or quarters do not receive antibiotherapy” (L16-18)
- L-37 and following: Reorder the Introduction. A study of 2019 showed how pathogens… (Maybe delete. It contributes nothing to the objective of the review). Add the reference here, at the beginning of the sentence.
- AU: Thank you, introduction has been reordered according to you suggestion. Reference has been added at the beginning of the sentence. (L39-124)
- L-55. Define what is CNS the first time you cite.
- AU: We apologize for the oversight, definition of coagulase-negative staphylococci (CNS) has been added (L92)
- L-277. Change for: “The ability of EC to detect subclinical mastitis is still controversial and further studies are needed, even if the association with other parameters can significantly improve predictability.”
- AU: Thank you, we have changed for “The ability of EC to detect subclinical mastitis is still controversial, and further studies are needed, even if the association with other parameters significantly improve predictability.” (L356-358)
- L-290. “Moreover, the possibility to identify subclinical mastitis at the dry off would be very important in order to perform SDCT.” I don't understand. How can clinical mastitis be diagnosed on drying by CMT? I recommend rewriting again.
- AU: Thank you for the comment. Clinical mastitis shows local and/or systemic clinical sign, that can be promptly detected through examination. Nonetheless, we think that some cases, in which symptomatology might be less evident, can be more difficult to identify. So CMT might be useful in order to detect or confirm disease. Sentence has been changed as follows: “Moreover, the possibility to identify not only clinical, but also subclinical mastitis at the end of lactation, before the dry off, would be very important in order to perform SDCT.” (L378-380)
- L-332. Define what is SCM the first time you cite.
- AU: We apologize for the oversight, we have changed for “subclinical mastitis” (L420)
- L-281 and following: Add that the CMT test, in addition to the surfactant detergent, has a pH indicator, which can modify the color of the milk sample.
- AU: Thank you for the comment, according to Marshall, R.T.; Edmondson, J., E.; Steevens, B. Using the California Mastitis Test | MU Extension Available online: https://extension.missouri.edu/publications/g3653, paragraph has been detailed as follow:
“Moreover, the purple color that results from the test is generally more intense in samples from infected quarters. This is because such samples have a basic or alkaline pH. The deep purple color indicates abnormal composition of milk due to mammary gland infection.” (L368-371)
- L-342 and following: “As affirmed before, mastitis is an inflammatory process of the mammary gland, resulting…” The explanation is strange. It could be made easier by saying that it is an inflammatory process, that there is an increase in temperature and that it is detected by the device.
- AU: Thank you for the suggestion, we explained the concept as follow:
“Mastitis is an inflammatory process of the mammary gland, leading to the production of prostaglandins and consequently to the increase of body and udder surface temperature, which can be detected by the IRT” (L434-436)
- L-350. “In bovine medicine, IRT is already extensively used for heat detection and prediction 350 of ovulation in cows, detection and assessment of lameness, assessment of animal welfare, 351 and feed utilization efficiency”. I recommend changing this paragraph of place and placing it before the previous paragraph.
- AU: Thank you, this paragraph has been placed before the previous (L431-433)
Reviewer 3 Report
This review focuses on potential tests for mastitis which can be performed on farm. In general it is a useful addition to the literature. A quick search reveals that there are a number of other recently published reviews in this general area, although with a somewhat different focus.
The main limitation of this particular review is that it does not really cover the possibilities of measuring a number of proteins or immune-related genes in milk. The currently very brief mention of metabolomics at L177 onwards should therefore be updated. Whilst these methodologies are mainly currently laboratory based, rapid tests using immune based technologies for proteins and PCR tests for genes are developing all the time. These seem likely to turn into tests that can be performed on farm in the fairly near future. While this aspect need not be covered in detail, I feel that it should be mentioned. Proteins which appear reasonably diagnostic include enzymes (e.g. N-Acetyl -β-d-Glucosaminidase (Nagase), Lactate dehydrogenase) and a variety of antimicrobial peptides (e.g. milk amyloid A (M-SAA), haptoglobin (HP), cathelicidin (CATH), and lactoferrin (LF)). Potential references to consult include: Sharun et al. Advances in therapeutic and managemental approaches of bovine mastitis: a comprehensive review. Vet Q. 2021 Dec;41(1):107-136. doi: 10.1080/01652176.2021.1882713; Giagu et al. Milk proteins as mastitis markers in dairy ruminants - a systematic review. Vet Res Commun. 2022 Jun;46(2):329-351. doi: 10.1007/s11259-022-09901-y; Loy et al. Current and Emerging Diagnostic Approaches to Bacterial Diseases of Ruminants. Vet Clin North Am Food Anim Pract. 2023 Mar;39(1):93-114. doi: 10.1016/j.cvfa.2022.10.006.
Other comments
L13 Summary. This sentence is confusing as clinical mastitis will have clinical signs.
L48 Italicize Streptococcus agalactiae
L55 CNS has not been defined
L65 omit comma
L90 change to “in the dairy cow”
L104 unclear what “ones” refers to. Rephrase.
L194 change to “.. of the following”
Section 3.1 would benefit from the addition of more scan images to compare normal with abnormal glands and teat walls. These should be labelled.
L236-242. None of these suggestions for the lower blood glucose in association with mastitis seem very likely as the body supplies glucose to the mammary gland rather than the other way around as suggested here. There is an extensive literature showing that cows in greater negative energy balance, in which glucose levels are reduced, are more susceptible to contracting mastitis. This is probably because combatting an infectious disease is energetically demanding. All of this involves functional changes to both the liver and the circulating immune cell populations. The details are outwith the scope of this review but this section should be revised.
L252 Destroyed not destructed
Figure 2 needs labelling.
The list of corrections to the English included above is by no means comprehensive. Although the paper is generally clear, there are nevertheless many places where use of English is not quite correct so this needs checking throughout.
Author Response
In accordance with your suggestions, we modified the article as follows:
- This review focuses on potential tests for mastitis which can be performed on farm. In general it is a useful addition to the literature. A quick search reveals that there are a number of other recently published reviews in this general area, although with a somewhat different focus. The main limitation of this particular review is that it does not really cover the possibilities of measuring a number of proteins or immune-related genes in milk. The currently very brief mention of metabolomics at L177 onwards should therefore be updated. Whilst these methodologies are mainly currently laboratory based, rapid tests using immune based technologies for proteins and PCR tests for genes are developing all the time. These seem likely to turn into tests that can be performed on farm in the fairly near future. While this aspect need not be covered in detail, I feel that it should be mentioned. Proteins which appear reasonably diagnostic include enzymes (e.g. N-Acetyl -β-d-Glucosaminidase (Nagase), Lactate dehydrogenase) and a variety of antimicrobial peptides (e.g. milk amyloid A (M-SAA), haptoglobin (HP), cathelicidin (CATH), and lactoferrin (LF)).
- AU: Thank you very much for the comment and the suggestion. Several studies are focusing their attention to find new reliable biomarkers. Metabolomics of course represents one of the recent and innovative techniques that might be useful to obtain successful results. The new methodologies focused on the evaluation of proteins and genes are very interesting to expound, thank you for the advice. These topics have been explained at L155-177 according to these references, some of which recommended by you:
- Hu, H.; Fang, Z.; Mu, T.; Wang, Z.; Ma, Y.; Ma, Y. Application of Metabolomics in Diagnosis of Cow Mastitis: A Review. Front Vet Sci 2021, 8, 1163, doi:10.3389/FVETS.2021.747519/BIBTEX.
-Sundekilde, U.K.; Poulsen, N.A.; Larsen, L.B.; Bertram, H.C. Nuclear Magnetic Resonance Metabonomics Reveals Strong Association between Milk Metabolites and Somatic Cell Count in Bovine Milk. J Dairy Sci 2013, 96, 290–299, doi:10.3168/JDS.2012-5819.
-Loy, J.D.; Clawson, M.L.; Adkins, P.R.F.; Middleton, J.R. Current and Emerging Diagnostic Approaches to Bacterial Diseases of Ruminants. Vet Clin North Am Food Anim Pract 2023, 39, 93–114, doi:10.1016/J.CVFA.2022.10.006.
-Sharun, K.; Dhama, K.; Tiwari, R.; Bashir Gugjoo, M.; Iqbal Yatoo, M.; Kumar Patel, S.; Pathak, M.; Karthik, K.; Kumar Khurana, S.; Singh, R.; et al. Advances in Therapeutic and Managemental Approaches of Bovine Mastitis: A Comprehensive Review. Veterinary Quarterly 2021, 41, 107–136, doi:10.1080/01652176.2021.1882713.
-Giagu, A.; Penati, M.; Traini, S.; Dore, · Simone; Addis, M.F. Milk Proteins as Mastitis Markers in Dairy Ruminants - a Systematic Review. Vet Res Commun 2022, 1, 329–351, doi:10.1007/s11259-022-09901-y.
-Adkins, P.R.F.; Middleton, J.R. Methods for Diagnosing Mastitis. Veterinary Clinics of North America: Food Animal Practice 2018, 34, 479–491, doi:10.1016/j.cvfa.2018.07.003.
The new paragraphs are reported below:
“Some interesting findings has been highlighted in two studies. The first has applied gas chromatography-mass spectrometry (GC-MS) on blood founding as reliable subclinical mastitis markers valine, serine, tyrosine, and phenylalanine 4–8 weeks pre-partum and valine, isoleucine, serine, and proline 4-8 weeks post-partum. The second study has applied Nuclear Magnetic Resonance Metabolomics (NMR) on milk founding that lactate, acetate, BHBA, butyrate, and isoleucine were in a higher concentration in high SCC samples. On the other hand, for the same samples, lactose, hippurate, and fumarate were at lower levels than in milk with low SCC levels.
Also, recent molecular techniques such as MALDI-TOF and commercially available quantitative real time PCR (rtPCR) tests are getting more common. MALDI-TOF is used to identify isolated organisms to genus and/or species level with a lower incidence of misclassification errors compared to traditional phenotypic speciation methods for some mastitis pathogens. Commercially available rtPCR tests reveal the advantages of being directly applied to milk samples containing preservatives: this procedure not only allows storage and shipment at room temperature, but also enables the identification of organisms that are difficult to culture in a timely fashion.
Another group of widely investigated potential biomarkers are Acute Phase Proteins (APPs), already commonly employed as clinical biomarkers of inflammation in serum, such as serum amyloid A (M-SAA) and haptoglobin (HP). Other proteins indicated as suitable mastitis markers are lactoferrin (LF) and cathelicidins (CATH). Some enzymes, such as Lactate dehydrogenase (LDH) and N-Acetyl -β-d-Glucosaminidase (NAGase) also seems to be reliable indicators of mastitis.”
- L13 Summary. This sentence is confusing as clinical mastitis will have clinical signs.
- AU: Thank you for the comment, the sentence has been changed as follow “The most prevalent form of the disease is subclinical mastitis, which leads, in absence of clinical signs, to decreased milk production, increased somatic cell count, and increased risk of clinical mastitis during lactation” (L11-12)
- L48 Italicize Streptococcus agalactiae
- AU: Done, thank you (L84-85)
- L55 CNS has not been defined
- AU: We apologize for the oversight, definition of coagulase-negative staphylococci (CNS) has been added (L92)
- L65 omit comma
- AU: Done, thank you (L81)
- L90 change to “in the dairy cow”
- AU: Changed, thank you (L121)
- L104 unclear what “ones” refers to. Rephrase.
- AU: Thank you for the suggestion, sentence has been rephrased as follows: “adopting good managemental practices to reduce the chance of transmission of the infection from sick to healthy quarters.” (L129-130)
- L194 change to “of the following”
- AU: Done, thank you (L262)
- Section 3.1 would benefit from the addition of more scan images to compare normal with abnormal glands and teat walls. These should be labelled.
- AU: Thank you for the suggestion, the possibly of see a comparison between an healthy and an infected quarter is very interesting and useful for the understanding of the description. We added one more scan image of a quarter infected by coli: “An example of quarter with E. coli infection can be seen in Figure 2 which shows the presence of hyperechoic spots and irregular aspect of the parenchyma.
Figure 2. Mammary ultrasound evaluation with longitudinal scanning (B mode) of a quarter with E.coli infection. Frequency: 6,5 Mhz. Depth: 150 mm. Focus: 50 mm. Presence of multiple hyperechoic spots and irregular parenchyma.” (L266-271)
- L236-242. None of these suggestions for the lower blood glucose in association with mastitis seem very likely as the body supplies glucose to the mammary gland rather than the other way around as suggested here. There is an extensive literature showing that cows in greater negative energy balance, in which glucose levels are reduced, are more susceptible to contracting mastitis. This is probably because combatting an infectious disease is energetically demanding. All of this involves functional changes to both the liver and the circulating immune cell populations. The details are outwith the scope of this review but this section should be revised.
- AU: Thank you very much for the comment. We really appreciate the statement you made, it’s very interesting for us. It represents a very useful food for thought. We found these other two theories trying to explain low blood glucose, even if there is an extensive literature concerning negative energy balance. According to your suggestion we improved this topic at L311-321 according to these references:
-Zhang, F.; Nan, X.; Wang, H.; Zhao, Y.; Guo, Y.; Xiong, B. Effects of Propylene Glycol on Negative Energy Balance of Postpartum Dairy Cows. Animals 2020, Vol. 10, Page 1526 2020, 10, 1526, doi:10.3390/ANI10091526.
-Gross, J.; Van Dorland, H.A.; Bruckmaier, R.M.; Schwarz, F.J. Performance and Metabolic Profile of Dairy Cows during a Lactational and Deliberately Induced Negative Energy Balance with Subsequent Realimentation. J Dairy Sci 2011, 94, 1820–1830, doi:10.3168/jds.2010-3707.
The paragraph has been improved as follows:
“Finally, a study highlighted a decrease in glucose blood concentration in case of clinical mastitis. Several authors explain this alteration with the concurrent presence of negative energy balance (NEB), very frequent in the transition period. Such NEB leads to significantly reduced serum glucose levels, thus affecting the energy supply of the immune system and predisposing animals also to mastitis onset. Few other hypotheses have been made. For example, another possible explanation is that blood flows to the infected udder and result in a limited supply of blood sugar secreted by the cell. Another hypothesis is that glucose is transported from the milk to the extracellular pathway to maintain the osmotic balance between the extracellular environment and the milk with the increase of Na and chlorine (Cl) when the body suffers from mastitis. In addition, hypoglycemia may be due to the accumulation of cells at the site of infection, which depletes the glucose in the local area.”
- L252 Destroyed not destructed
- AU: Done, thank you (L331)
- Figure 2 needs labelling.
- AU: Thank you for the comment, figure 3 has been detailed as follows: “Figure 3. Infra-Red Thermography of affected quarter. The presence of red color in correspondence with the teats and orifices indicates increased surface temperature. No relevant difference between udder and teat temperature is detected, which is present in healthy quarters.” (L437-440)
- The list of corrections to the English included above is by no means comprehensive. Although the paper is generally clear, there are nevertheless many places where use of English is not quite correct so this needs checking throughout.
- Thank you for the comment, we have checked English throughout the article. We hope language will be now more appropriate.